# Weight stigma after bariatric surgery: A qualitative study with Brazilian women

Mariana Dimitrov Ulian[1]*, Ramiro Fernandez Unsain[1], Ruth Rocha Franco[2‡], Marco Aurélio Santo[3‡], Alexandra Brewis[4], Sarah Trainer[4], Cindi SturtzSreetharan[4], Amber Wutich[4], Bruno Gualano[5], Fernanda Baeza Scagliusi[1]

1 Departamento de Nutrição, Faculdade de Saúde Pública da Universidade de São Paulo, São Paulo, SP, Brasil, 2 Unidade de Endocrinologia Pediátrica do Instituto da Criança do Hospital das Clínicas da Faculdade de Medicina da Universidade de São Paulo, São Paulo, SP, Brasil, 3 Divisão de Cirurgia do Aparelho Digestivo, Unidade de Cirurgia Bariátrica e Metabólica, São Paulo, SP, Brasil, 4 School of Human Evolution and Social Change, Arizona State University, Tempe, AZ, United States of America, 5 Grupo de Pesquisa em Fisiologia Aplicada e Nutrição, Faculdade de Medicina da Universidade de São Paulo, São Paulo, SP, Brasil

☺ These authors contributed equally to this work.
‡ RRF and MAS also contributed equally to this work.
* mari_dimi@hotmail.com

**Data Availability Statement:** The data relevant to this paper are available from figshare, https://doi.org/10.6084/m9.figshare.23556324.v1.

## Abstract

Prior studies suggest that one anticipated benefit of bariatric surgery is the achievement of a thinner body, one that is less subject to perceived negative judgment and condemnation by others. However, additional analyses also indicate that stigma may persist even with significant post-surgery weight loss. To investigate the stigma-related perceptions and experiences of women who have undergone bariatric surgery and the resulting body transformations, we conducted individual, semi-structured interviews with thirty Brazilian women (15 aged 33–59 and 15 aged 63–72). The resulting text was then analyzed using thematic analysis. We found that some form of weight stigma persisted for our participants, regardless of weight loss. Ongoing experiences of stigma were also evidenced by the constant internal and external vigilance reported by the women, as well as their articulated efforts to distance themselves from their previous bodies. Additionally, participants reported being judged for choosing an "easy way out" to lose weight. Those in the older group reported that weight stigma was entangled with ageism: older participants received mixed messages underscoring the ways that weight and age may interact in doubly stigmatizing ways. Family and close peers were especially powerful sources of stigma experiences. Collectively, these results show that weight stigma persists even when people undergo a procedure to lose substantive weight and that the degree and types of stigma experiences are influenced by gender and age. Our study suggest future research should explore whether a targeted approach might be more effective, for example, an approach that would emphasize the importance of developing coping strategies with respect to experiences of stigma and discrimination after surgery.

**Funding:** This work was supported by the São Paulo Research Foundation (FAPESP), grant number 2019/00031-0, granted to Mariana Dimitrov Ulian. In addition, Fernanda Baeza Scagliusi received grant by FAPESP (grant number 2017/17424-9) and CNPq (grant number 309514/2018-5). The funders had no role in study design, data collection and analysis, decision to publish, or preparation of the manuscript.

**Competing interests:** The authors have declared that no competing interests exist.

## Introduction

One of the most effective means currently available to individuals seeking to lose weight is bariatric surgery. Bariatric surgery, also known as weight-loss surgery, refers to a number of different procedures, all of which aim to induce severe weight loss through the surgical reduction of the stomach and/or intestines (e.g., [1–4]). Bariatric patients often have a history of complex anxieties and negative social experiences stemming from weight stigma [5–8]. The weight loss that accompanies the surgery, therefore, has been generally related to positive changes in body image and social interactions [9, 10].

In Brazil, bariatric procedures have been available at no cost to patients since 1999 through the Unified Public Health System or the Sistema Único de Saúde [11]. In this respect, Brazil parallels certain European countries that have also made bariatric surgeries widely available to citizens within an integrated healthcare system. Brazil is, however, economically, socially, and culturally quite distinct from these other nations. The sociocultural implications of the surgery, including experiences around weight stigma, have been understudied in a Brazilian context. For example, despite the highly-publicized increase in the number of bariatric surgeries performed within the Brazilian public health network, many experts still consider this number insufficient, as only about 8% of all applications for surgery are approved and financed [12]. Also, there are regional inequalities, with the wealthier South and Southeast regions reporting the most bariatric surgeries, and the North and Midwest regions reporting the least [13].

Stigma can be defined as negative judgments and behaviors aimed at a devalued attribute or condition that disqualifies a person or family or community with this attribute or condition from full social acceptance in a particular cultural context. Stigma results from a cascade of social interactions, in which the individual (or family or community) exhibiting the stigmatized trait is distanced from others, all of whom supposedly adhere to a normative standard. This in turn legitimizes various social and economic discriminations and exclusions [14]. Weight stigma is the process by which people with higher body weights are socially classified as less valuable or desirable because of their weight [15]. Following Brewis [16], we define weight stigma as moral discrediting that people experience from others or apply to themselves because of the negative social meanings attached to the size of bodies. Stigma related to very large bodies tends to be especially pernicious. Weight stigma has been identified as an important barrier to obtaining consistent healthcare services [17] and has significant emotional and social impacts for people who are identified as obese [15, 17–21].

People with higher body weights often internalize weight stigma and end up agreeing with dominant social messages stating that the adverse judgments of them based on their weight have merit [15, 21, 22]. It is also common that people who have been labeled as "obese," "overweight," and/or "fat" feel individually responsible for making specific choices around diet and exercise to achieve a more socially acceptable body, i.e., a thinner body. This idea that individuals should make "good choices" is centered in powerful sociocultural notions around the responsible management of the physical body and in the social science literature has been interpreted as a form of moral biocitizenship [23, 24].

Some previous research, including research within Brazilian populations, has indicated that people with a higher body weight consider bariatric surgery a means to achieve a thinner body, and one that is less subject to the negative judgments of others [25, 26]. Nonetheless, studies also suggest that stigma persists in different forms after bariatric surgery. This persistence may involve negative judgments regarding the decision to do such a procedure and/or of the post-surgical body, and may therefore influence the candidate's post-surgical dietary adherence and overall well-being [4, 27–29]. For example, one study [6] showed that participants who underwent bariatric surgery often heard from their familial and social circles that bariatric surgery

was an "easier" method of weight loss in comparison to traditional methods centered on diet and exercise. These traditional methods are often framed as involving moral biocitizenship and "hard work" done upon and with one's physical body. Doing something the "easy way" was therefore equated with "cheating" at weight loss and thus was a mechanism of stigma creation.

Other research has shown profound shifts in people's lived experiences in the years after bariatric surgery, mirroring their bodily transformations [30–33]. Negative judgments of post-bariatric bodies are commonly reported in the literature, often centering on the persistence of side effects from the surgeries [34–38] and/or difficulties with managing the rigid personal habits the surgeries necessitate [32–38]. Other negative judgments stem from the shapes and sizes of the bodies produced by the surgery and its accompanying rapid weight loss. Weight loss induced by surgical intervention is so rapid, for example, that it often produces a great deal of excess skin. Furthermore, people tend to lose weight in their heads, arms, and legs before they lose it in their torsos [4]. This pattern of weight loss and loose skin does not always conform to societal body shape ideals, especially for women.

Little is currently known about how women at different life stages (i.e., adulthood vs. elder-hood) deal with the profound body transformations caused by the surgery and attendant weight stigma or whether life stage affects the ways in which women assume personal responsibility for maintaining their modified bodies after bariatric surgery. From the more generalized body image literature, we know that body dissatisfaction tends to be fairly stable or to decrease across the life span [39]. However, most studies have not focused on those who lose weight but rather on those who gain weight as they age [40]. Some studies, mostly based in the US and Europe, have shown that older women experience increased body concern as they grapple with feelings of being "trapped" in aging bodies [41]. Higher body weights (past or present) may therefore interact with age and experiences of weight stigma and negatively impact well-being in women's later years. As yet, there is little study on this specific point.

Finally, there is a dearth of research on bariatric surgery and other extreme weight loss experiences in the Global South. Brazil is an interesting location to conduct research in this area of study for a number of reasons, including the fact that sociocultural messages about the "beautiful body"–characterized in women by a thin waist but big breasts and bottoms–are extremely powerful (e.g., [42]). Moreover, the universal healthcare system is offset by extreme socioeconomic inequities, and attitudes about women (both in terms of their relationships with their families and as citizens of the Brazilian state) are undergoing rapid transformation [43].

Therefore, in this study, we aimed to qualitatively understand how weight stigma has affected Brazilian women who have undergone bariatric surgery. Beyond its unique focus on Brazil, our study's novel approach centers on an age-based description of older and younger women after bariatric surgery, with a particular focus on older women since their experiences are not well documented in the literature. We also were interested in understanding how the participants perceived and internalized (or not) weight and other types of stigmas before and after surgery and how they dealt with the bodily and eating-based transformations brought on by the surgery. Our study can suggest whether a targeted approach might be more effective, for example, an approach that would emphasize the importance of developing coping strategies with respect to experiences of stigma and discrimination after surgery.

## Methods

### Research setting and study population

The Ethics Committee at the School of Public Health of the University of São Paulo and the Hospital das Clínicas of the School of Medicine of the University of São Paulo (HCFMUSP)

approved this study (Approvals 4.031.373 and 4.143.745, respectively). Participants provided both digitally recorded oral consent and digitally recorded informed consent, and both processes were also approved by the Ethics Committee. All research procedures adhered to the regulations in the Declaration of Helsinki as revised in 2008.

This research was carried out at HCFMUSP. This institution is a tertiary, referral-based teaching hospital located in the most populated city in Brazil: São Paulo. To be eligible for bariatric surgery in this institution, patients must have a medical recommendation provided by a private or public clinic. Then, the patient has a trial session with the physicians of HCFMUSP, who evaluate the patient to see if they meet the criteria to be included on the waiting list for bariatric surgery. If successfully placed on the waiting list, the patient then has annual appointments with a surgeon and receives nutritional and psychotherapeutic support. The appointments continue until the surgery is performed, and the wait for surgery can last years. After the surgery, the appointments with the physician are initially frequent, but if the patient appears to be recovering well, the visits become more spaced out. Nutritionists and psychotherapists also provide follow-up care, and, if deemed medically necessary, the patient can be referred to other specialties (e.g., psychiatry).

For this study, we used a purposeful sampling method focused on women who had bariatric surgery at HCFMUSP between one and five years prior to our meeting. Before the onset of the COVID-19 pandemic between November 2019 and March 2020, we directly recruited women who underwent bariatric surgery by visiting the clinic and interacting with post-operative patients. Interested participants provided Author 1 with their names and contact information. During the early months of the pandemic (June 2020 to August 2020), post-operative patients already being followed by the Bariatric Surgery Outpatient Clinics of HCFMUSP were also recruited by us, as the clinic provided a list of names and contact numbers of eligible participants. Forty-six potential participants–whose contact information we either gathered directly pre-Pandemic or was provided to us by the clinic during the Pandemic–were contacted via a WhatsApp message by Author 1. We chose this application because it was the easiest of the secure tools available to participants for facilitating a remote interview. Also, the app's end-to-end encryption ensured that information was only exchanged between the researcher and the participants, with no access available to any third party.

In the initial contact, Author 1 introduced herself, explained why she was contacting them, the purpose of the research, and clarified that if they agreed to participate, the participation would be voluntary, and the findings would be kept confidential. This study does not qualify as Patient and Public Involvement (PPI) research, because PPI refers to an active partnership between members of the public and researchers, in which members of the public work alongside the research team and are actively involved in contributing to the research process as advisers and possibly as co-researchers [44]. This did not happen in our study.

Thirty women agreed to participate after initial contact by Author 1. The methodological framework of the present study focuses on two distinct age categories. The sample consisted of 30 women who we categorized as "younger" (aged 33–59 years, n = 15) or "older" (aged 63–72 years, n = 15) at the time they underwent bariatric surgery. These sample sizes meet or exceed empirically-based guidelines for achieving thematic saturation in qualitative research [45].

## Data collection

Author 1 conducted semi-structured interviews with each of the women who consented to be in the study. Given issues with the proposed length of the initial interview protocols (which averaged three hours), participants subsequently were offered several shorter interviews spread out over time, rather than a single, long session. For most of the participants, two interviews

were sufficient to address the interview protocols and prompts, but, in several cases, a third interview was necessary. Author 1 has extensive experience with semi-structured interviews, having conducted this type of research during her M.A. and Ph.D. work. The interview protocol that guided the interviews is presented in S1 File, and the interview schedule is presented in S2 File.

The 2–3 interviews per woman were conducted in the same or consecutive weeks of each other. Pandemic interviews conducted remotely were audio-recorded and then transcribed verbatim by an external company. Non-verbal behaviors such as gestures were observed and recorded in separate handwritten notes. These procedures meet qualitative criteria of methodological rigor [46]. This divided-encounters approach proved more practical and feasible for the participants' schedules. Notably, the time lapse between the first and second (and, when necessary, third) interviews increased data quality, since it gave Author 1 an opportunity to review recordings of the first encounters and to note gaps that needed more attention in subsequent sessions.

## Data analysis

Participants' life story narratives were assembled from the data collected in the interviews so that we could investigate how interviewees perceived and internalized weight and other types of stigmas before and after bariatric surgery and how they dealt with bodily and eating transformations brought on by the surgery. The interviews were transcribed and then analyzed in Brazilian Portuguese. Here, excerpts that are relevant to the analytic narrative have been translated to English, but we have also retained and provided the original Brazilian Portuguese.

Following Braun and Clarke [47], thematic analysis comprised a series of phases. Initially, Author 1 familiarized herself with the data set by reading and rereading the transcripts and noting down initial analytical observations. Then, Author 1 coded the transcripts, systematically identifying and labelling relevant features of the data in relation to the research questions and grouping together similar data segments. Afterward, we moved on to the development of themes and in this stage of the process, Author 1 worked closely with the other authors. As Braun and Clarke [47] highlight, themes are not simply sitting in the data waiting to be uncovered. Rather, Author 1 clustered together codes to create a plausible mapping of key patterns in the data. The themes were then named, categorized, and organized and the final product then provided a road map for the write-up. Finally, Author 1 presented the analytic narrative with vivid and compelling data extracts.

## Results

All thirty women who participated in this study underwent bariatric surgeries between 2016 and 2019, before the onset of the COVID-19 pandemic. We provide an overview of the general characteristics of our participant sample in the table below (Table 1). This information was obtained directly from the participants during the individual interviews (see S1 File).

Overall, we found that after bariatric surgery and its accompanying weight loss, our participants reported that they experienced higher social acceptance in their daily interactions and less discrimination and overt oppression, but they also said that they still faced internalized and reproduced weight stigma. In addition, they assumed specific body-centered attitudes and dealt with conflicting messages regarding their bodies after the surgery. Their interviews yielded four themes regarding these experiences, detailed below, with particular attention paid to the experiences of the older women. All the excerpts that are italicized and have quotation marks around them are English translations and/or original Brazilian Portuguese quotations from the interviews. We could not include all of the original Portuguese quotes due to length

**Table 1. General characteristics of women participating in the study.** São Paulo, Brazil, 2020.

| | Younger women (n = 15) | Older women (n = 15) |
|---|---|---|
| **Current age** (yr.), mean | 45 | 67 |
| **Age at surgery** (yr.), mean | 43 | 64 |
| **Anthropometry** | | |
| Current body mass index (kg/m$^2$), mean | 31.2 | 33.9 |
| Body mass index (kg/m$^2$) at surgery, mean | 44.9 | 44.8 |
| Highest body mass index (kg/m$^2$) pre-surgery, mean | 51.9 | 51.9 |
| Lowest body mass index (kg/m$^2$) post-surgery, mean | 29.5 | 31.5 |
| **Self-perceived skin color**, n | | |
| White | 4 | 11 |
| Black | 1 | 2 |
| Brown | 10 | 2 |
| **Relationship status**, n | | |
| Single | 3 | 1 |
| Married | 8 | 6 |
| Common-law marriage | 1 | 1 |
| Divorced | 2 | 3 |
| Widowed | 1 | 4 |
| **Education**, n | | |
| Incomplete elementary school graduation | 0 | 5 |
| Graduated from elementary school | 2 | 2 |
| Incomplete high school graduation | 1 | 1 |
| Graduated from high school | 5 | 4 |
| Incomplete college graduation | 0 | 2 |
| Graduated from college | 6 | 1 |
| Postgraduate-level studies | 1 | 0 |
| **Household conformation**, n | | |
| Lives alone | 2 | 4 |
| Lives with only one family member | 2 | 7 |
| Lives with two family members | 5 | 4 |
| Lives with three or more family members | 6 | 0 |
| **Monthly family income** (value in U.S. Dollars), n | | |
| ≤ 397.00 | 6 | 8 |
| 397.01–794.00 | 4 | 6 |
| 794.01–1,986.00 | 4 | 1 |
| 1,986.01–3,972.00 | 1 | 0 |

limitations, but we consider the inclusion of those highlighting key sociocultural concepts and phrases of vital importance in understanding the participants' points of view.

## Theme 1: Undergoing bariatric surgery means others will judge me

**Common experiences among the younger and older participants.** Participants routinely mentioned receiving stigmatizing judgments for undergoing bariatric surgery. These judgments came from family members, coworkers, neighbors, healthcare professionals, and strangers. Judgmental statements the women reported to Author 1 included being told that they *"took the easy way out"* (*"você foi pelo caminho mais fácil"*), *"gave up easily"* (*"você desistiu fácil mesmo"*), and *"should have shut their mouths [and just stopped eating]"* (*"você deveria ter fechado sua boca"*). Participants reported that they were also told they were *"cowards"* for not

exercising more self-discipline and *"crazy"* for undergoing an elective surgical procedure. It was common for our participants to report that they were told by others that they could have lost weight if they had tried other strategies, specifically dieting and food restriction.

Far from the surgery being an "easy way out", our participants repeatedly said it was the only choice possible and made after exhausting all other options. Before choosing this path, they reported spending years undergoing weight-loss treatments with different healthcare professionals but that these did not result in sustainable changes to their weight. One participant (a younger woman), responding to accusations that she should have relied on dieting, commented, *"I have been on a diet since I was three years old. Today I am 34. What would make me achieve it now if I could not lose weight during this period?. . . Surgery was the most viable method I had at the time to keep myself a healthy person."* Other participants reinforced the sentiment that undergoing bariatric surgery is not easy.

Other judgments participants often reported hearing centered on the idea that they only lost weight in the short-term because they had bariatric surgery and as a result, they would be unable to maintain the weight loss. Sonia (a younger woman), for example, mentioned that her sister had started to gain weight. She remembered that their mother suggested the sister ask advice of Sonia about eating and exercise. Sonia's sister replied, *"Mom, she only lost weight because she had bariatric surgery; otherwise, she would have been 'fat' all the same. Moreover, soon she will gain weight again, do you think she will stay like this? This is not forever, no."*

**The experiences of the older participants.** Among the older women participants, other judgments were also consistently reported. This group reported, for example, being judged for undergoing the surgery at their (advanced) ages. Maddalena recalled that her son told her that the surgery was an *"aggression to the body"* (*"é uma agressão para o corpo"*) and questioned the value of doing it. The older women commented that some of their relatives said they did not want them to do the surgery because they feared the associated health risks for their age cohort. Other relatives voiced concerns that the side effects of the surgery were worse than being "fat", especially at the participants' older age. Eugenia, for instance, shared that she heard from others that she *"was better before having the surgery,"* because sometimes she experienced side effects, such as vomiting, even though Eugenia herself emphatically said she felt better post-surgery. Rita reported that when she told family members she could not eat all the food on her plate, her niece said that if it was to be like that (i.e., with restrictions on eating), it would have been better not to have the surgery. Once again, Rita herself felt that having the surgery was for the best.

## Theme 2: Constant monitoring after surgery must be "accepted" so I don't become a "monster" again

**Common experiences among the younger and older participants.** Post-surgery, participants reported a great deal of surveillance and monitoring. This monitoring was supposed to help maintain the surgery results, i.e., the weight loss. Women reported that monitoring–in the form of visual oversight, food management, and verbal feedback–came from family members, coworkers, clients, health professionals, and strangers. They also reported self-monitoring.

Common comments that the participants received from others stressed that the surgery was only a support and that the participants must do "everything right" to not gain weight again. Sonia (a younger woman) said that after the surgery, it was necessary to adapt and change herself in order to *"not go back to be[ing] 'fat', to be[ing] a monster, like they [her family] used to say"* (*"para não voltar a ser gorda, a ser um monstro, como eles [a família dela] falavam"*). Participants told us that they saw the monitoring from others in a positive light.

Women also reported that they monitored, and sometimes felt critical of, their own eating. In general, the participants said that even if they felt the desire to eat, they avoided food. Therefore, participants expressed chronic concerns about self-monitoring and behaving "correctly" in relation to eating. During times when they could not sustain this "correct" eating, however, women expressed concern that they should be doing more to improve upon the surgical results. Cristina (a younger woman) said she felt that when she ate more than she was supposed to, *"it is as if I were not taking this seriously, I am ashamed of myself sometimes."* Another participant, Myrian, remarked, *"I joke sometimes, saying, 'I am obese, take it [some food] away from me because I am obese'. . . Some people are addicted to cigarettes, I am addicted to food. So if you do not have consciousness about who you are, you will return to what you are."* Other participants shared a similar perception.

Participants reported that medical professionals also seemed to extensively monitor them and their weight, as illustrated by the experience of Roberta (an older woman). She said that she went to an appointment at the health center and after the doctor assessed her weight and noted that Roberta had gained two kilograms, he got mad and said, *"You will lose the [gains of the] surgery; there is nothing else to be done, what is done is done. If you do not lose more weight, there is no point to return here."* When Author 1 asked how she felt about his reaction, Roberta answered, with a resigned tone, that she agreed with him. Similarly, many participants reported being reprimanded by nutritionists for weight gain or weight loss deemed insufficient. For example, Marcia (a younger woman) said, *"I had an appointment with [the nutritionist] once. She got mad at me, angry at what I ate. She said: 'You lost [only a] little weight; you were supposed to have lost more. [It is] because you are eating.' You know? She was rough, direct."* Another nutritionist told Rosana (a younger woman) to lose more weight, proposed changes to her daily eating, and recommended that she should eat food without broth or other liquids.

Participants consistently reported that they were monitored by many people in their community and social networks regarding their eating choices. Marcia (a younger women) said, for instance, *"There was a woman who told me: 'pasta is fattening',"* and then this woman advised Marcia not to eat pasta. Participants reported that they were frequently criticized if they did something that others perceived as jeopardizing their weight loss, such as eating more than they should or consuming what others considered to be "fattening foods.". Luanna (a younger woman) and Teresa (an older woman) shared a similar, very common narrative around this type of monitoring. Because of the surgery, they could only eat a small volume of food at any given time, so they divided their daily caloric consumption into small portions throughout the day. Their relatives were constantly surprised that they ate several times a day. Luanna said that when her husband saw her eating "again," he would censure her. She said that she became very angry with his attitude because she only ate when she was hungry, and when she checked her weight, it continued to decrease. Marina (an older woman) said that when she went to see her niece, the niece would put food on Marina's plate and say, *"That is it, that is what you will eat; you will not eat more."* Cristina (a younger woman) said that her sisters also "supervised" her eating, telling her to stop eating when they felt she had had enough food and telling her not to buy certain food items at the supermarket. She said she chose to see this support positively: *"They are always by my side"* (*"Elas estão sempre do meu lado"*).

Participants shared that relatives were particularly vocal in their monitoring. One articulated justification that underpinned familial monitoring of the post-operative women involved the complexity of the procedure, as illustrated by Bruna (a younger woman). She said she remembered hearing from her sisters, *"You take care. You had bariatric surgery, you had such a complicated surgery, [but now] you are gaining weight."*

**The experiences of the older participants.** In addition to being reminded about the complexity of the surgery, the older women participants also shared that their relatives highlighted how worried they were about them during the procedure itself, as well as in the years before the surgery when they suffered from a range of health problems and had difficulty performing daily activities. Teresa, for instance, mentioned that a relative told her, *"It is better to stay like this [with a higher body weight] than to die on a surgery table."* Older women consistently told Author 1 that relatives would constantly monitor them and voice concern over their choices. Marina said that her niece told her, *"If you return to the hospital [because you regained weight], I will not carry you, I will not!"* These admonitions demonstrated the worry that relatives had that the participants were in danger of returning to their previous conditions.

When we asked our participants how they responded to such admonitions, they told us that they agreed with these comments. As Teresa commented, *"They are right. I accept, and I listen to everything, they are right, and I am wrong."* In the interviews, women often voiced that they did not want to be a "burden" (*"ser um peso"*) to their relatives.

## Theme 3: Stigmatizing judgments of my bariatric body are everywhere

**Common experiences among the younger and older participants.** Commonly voiced judgements of participants took many forms. One judgment involved telling post-operative women that they were "too thin," and, usually, this observation was accompanied by the comment that continued weight loss would make them look bad. Luanna (a younger woman) reported a very typical reaction: *"Many people tell me, 'Do not lose more weight; otherwise, you will be ugly'"* (*"Muita gente fala: 'Não emagreça mais não, senão você vai ficar feia'"*). Other comments involved asking participants if they were sick because of the weight loss. Catia (a younger woman) reported, for instance, *"Sometimes they [neighbors, friends] think I am sick, right?"*

On the other hand, participants who described experiencing a small weight regain after their initial, sustained weight loss also described being reprimanded by others. For example, Eugenia (an older woman) said that after the surgery, she got *"really skeletal"* (*"eu fiquei esquelética"*), and she purposely regained four kilograms to *"return to a normal body"* (*"voltar ao corpo normal"*). This increase in her weight reduced the visibility of her loose skin and made her feel more comfortable. Nonetheless, she remembered hearing alarmed messages from family members, many of whom remarked that they were worried: *"Oh my, you are gaining weight!"* She said that this kind of criticism harmed her, lowering her self-esteem, but she also said she could not reply to these comments and just had to accept them because they stemmed from love and concern.

Similarly, participants who did not lose "enough" weight after surgery shared that they were often critiqued by family members who expected dramatic, visible weight-loss in the months post-procedure. Myrian (a younger woman) recalled that her parents questioned the efficacy of her surgery because, according to them, "everybody" who did the surgery is thin–except for Myrian, who still had a higher body weight. As Myrian remarked, *"It's like, 109 kg is still 'fat', you know? I am [still] obese [after the surgery]."* She pointed out that the doctor considered her surgery a success and told her that she was doing great because she had already lost a certain percentage of weight. Outside the clinic, however, social expectations that a bariatric surgery will produce a skinny self were very common and very powerful. Myrian, for instance, despite critiquing her relatives' opinions and having the support of her doctor, had mixed feelings about her progress. She said, *"Sometimes I really feel like I could have lost more weight. I feel it. I could be a lot thinner."*

**The experiences of the older participants.**   Among the older participants, it was common for them to hear that they had aged after the surgery. Arlete said that when she met her friend some months after the surgery, the friend was surprised: *"When she saw me, she was so amazed, and it shocked me, you know? [She said] 'I cannot believe it is you. You look like a ninety-year-old woman!'"* Arlete expressed being extremely upset with these comments but said, resigned, that now, *"there is no turning back."*

The older participants mentioned receiving specific comments about changes in their bodies. For example, Rita said that her friends told her to lose weight in her belly area because everything else was fine. Rita reported that she agreed that it would be perfect for her if she lost weight in the belly–but that this was not something she could control or direct. Carmen mentioned perceiving looks and hearing comments due to the excess skin in her arms after bariatric surgery. Most of the older participants stated that they did not respond to the comments involving their bodies or appearance.

### Theme 4: I judge other, "unsuccessful" bariatric patients

**Common experiences among the younger and older participants.**   Participants expressed judgments about other people who underwent bariatric surgery and gained weight after the procedure. Eunice (an older women) expressed that people who did the surgery had an opportunity to have a "good body" (*"um corpo bom"*) and, therefore, must maintain that good body post-surgery because *"The obese [person] is on the verge of death, there is no way out, there is no future for him."* Women reported that weight gain meant that a person did not understand the real purpose of the surgery: that it was not the final solution and that an individual still had to work to change eating habits. *"People think 'now that I did bariatric surgery, I can eat whatever I want,'"* said Marcia (a younger woman). Overeating and "letting their guard down" about eating were pointed out by our participants as explanations for why people gained weight after the surgery. Carmen (an older woman), for example, made the following observation about her colleague who gained weight post-surgery: *"He must have seen some delicacy that whetted his desire, and he could not control himself."*

Participants expressed that the success of the surgery depended on how seriously the patient took the procedure and how strictly they followed the healthcare professionals' recommendations. The women with whom we spoke also stressed that eating with "control" was an important aspect of the surgery's success. Having a "good mentality" (*"trabalhar a cabeça"*), "willpower" (*"força de vontade"*), and "discipline" (*"depende da disciplina"*) were all portrayed as virtues in the post-surgical individual. Working at "controlling themselves" (*"eu me controlo"*), "ignoring desires" (*"ignorar desejos"*), and "policing oneself regarding the amount of food and the junk-food eaten" (*"eu me policio da quantidade da porcaria que eu vou comer"*) were likewise portrayed as virtuous. For participants, weight gain in another person after bariatric surgery indicated that the person did not have these virtues. The weight gain was an outward manifestation of something being awry behaviorally–in other people.

In this theme, there were no differences between the younger and the older women narratives.

## Discussion

In this study, we aimed to understand the ways in which weight stigma affected a group of Brazilian women who underwent bariatric surgery. We were also interested in understanding how women at different life stages (i.e., adulthood vs. elderhood) dealt with the profound transformations caused by the surgery and attendant weight stigma.

Participants agreed that surgery itself was stigmatized, particularly because it was considered an "easy way out" to lose weight. This same theme of stigma related to surgery as "cheating" has also been observed in ethnographic studies of bariatric patients in the US and Canada [6, 48–50]. Similar perceptions have been documented within Australian [27, 51, 52], Norwegian [53], and English and Scottish [33, 54] populations. In these neoliberal contexts, being thin is seen as a personal responsibility [55], and our results suggests that this sample from Brazil seems to conform to this same theoretical framework. This was observed, for example, when different people suggested that participants did not try "hard enough", with their own personal resources, to meet slender norms, and instead had the bariatric surgery.

Although there is quantitative research on the health-related consequences of bariatric surgery in older patients [55, 56], there is a dearth of qualitative data on the experiences of this population. Our study provides some insight into the experiences of older women who undergo bariatric surgery. For example, our older participants received more mixed messages about their decision to undergo bariatric surgery–part of why they felt more stigma. When women started losing substantive weight, they reported that family members and friends wanted them to eat more and to return to their previous, larger sizes–but older women were more often told that weight loss made them look elderly. Studies of bariatric populations in other national contexts have documented similar communications from friends and family with patients (e.g., [4, 6]). What was novel about our findings is that the women interviewed, and especially the older women with whom we spoke, consistently expressed the strong belief that they just needed to accept negative feedback from their families and communities. This stated acceptance is different from the Brazilian study of Liebl et al. [57] with younger participants, where participants stressed that they did not allow negative, unsupportive people to influence their health decisions. This finding also differs from the findings of a study in the U. S. across age cohorts [4], whose participants similarly stressed that they did not listen to people who said negative things to them post-surgery. In our study, some of this stated acceptance possibly reflects dominant social mores and attitudes around women, and in particular older women [58]. Our study therefore underscores the complex connections between weight and age stigma in the Brazilian context [59].

The bodies of our participants after bariatric surgery were a specific source of stigmatizing messages from others. Our participants expressed an underlying feeling that they were always under evaluation to see if they would maintain the expected body weight (and, for the older participants, an acceptably aging appearance). Our results show that these Brazilian women faced serious obstacles to moving beyond the stigma of their weight–even after losing weight. Studies have shown that, in Brazil, the body that is desired is the one that is worked on, cared for, unmarked (e.g., without wrinkles, stretch marks, cellulite, blemishes), and one without excess "fat" [60–64]. Brazilian self-care through exercise, weight management, and cosmetic procedures (among other interventions) is seen as essential to modern Brazilian femininity [64]. In this context, a failure to produce the desired body even after a weight loss surgery potentially connects directly to individual moral failures, which may have the potential to lead to negative mental health outcomes.

Stevens [65] points out that the symbolic "fat" body, which was viewed as "out of control" before bariatric surgery, ideally becomes a slimmer and more disciplined body afterwards. This transformation is understood to be ensured by strict adherence to a particular lifestyle and to disciplinary tools (e.g., physiological limitations and dietary restrictions) that distanced people from the formerly higher-weight, stigmatized body. Participants repeatedly reported feeling the necessity to demonstrate to other social actors (like friends and family) their ongoing commitment to weight loss and to eating with "control." Similarly, one participant shared that the surgery gave them a chance at a "good" body: a disciplined, non-obese, not-"fat" body.

We suggest that our participants understood weight gain to be a return to a "bad" body, since a "bad" body is indicative of a bad person, and a moral failure in this context. As we noted at the outset, this idea that individuals should make "good choices" is centered on the requirement of responsible management of the physical body in order to be a functioning, moral biocitizen [23, 24]. Among our older participants, this seemed to be entangled further with experiences of what was interpreted as ageism: a stereotypical and often negative view of older people as less competent [66]. This view exacerbated attempts by others to monitor the older women.

Our participants, especially older women, commonly stated that external warnings, admonitions, restrictions, and advice needed to be tolerated and were even warranted. This may suggest an internalization of certain kinds of stigma by these women–that they believed the stereotypes stating that because they had a higher body weight and/or because they had bariatric surgery and/or because they were older, they were somehow less disciplined and less competent and required supervision by others. Alternatively, it could reflect women's belief in certain age-related and feminized norms, i.e., one must accept one's family's advice and dictates. Illustratively, studies have shown that Brazilian culture encourages acceptance of a situation, as well as lack of inquiry into its causes [67].

In a U.S. study, the authors stressed the importance of the bariatric program's pre-operative educational classes and post-operative support group in patient long-term success. These classes and groups provided participants with a framework and preparation for understanding the many reactions that their social and work circles might have after bariatric surgery [4]. Most importantly, perhaps, the classes also encouraged challenging situations if they did not enhance wellbeing post-surgery. Although our participants also had pre- and post-operative support groups, this support may have been insufficient to combat stressful social messaging, especially if the support focused exclusively on clinical questions and issues.

Our participants also expressed stigmatizing views regarding other people who gained weight after surgery. For them, those who gained weight must not have been good biocitizens, since biocitizens should be able to demonstrate vigilant and controlled eating. In making these judgments, participants reproduced a moral biocitizenship [23, 24] themselves, as they discussed peers' mistakes and habits as not-quite-compliant-enough post-bariatric citizens. They internalized that the right way to live after bariatric surgery is to have "willpower," "control," and a "good mentality." Their narratives showed that they tried their best to put this thinking into daily practice, even as their virtuous efforts were called into question when they themselves gained weight or found themselves unable to sustain weight loss. Keeping this articulated perspective (willpower-control-good mentality) at the forefront made participants feel that they were genuinely compliant with what was necessary to succeed after bariatric surgery. This attitude, as reported to us, differentiated them from the "others" (i.e., the "unsuccessful" bariatric colleagues) and resulted in our participants producing themselves as normative subject-citizen-bodies. Participants in the U.S. showed this judgement as well [4]. These attitudes suggest that people who have undergone bariatric surgery, even across different sociocultural or clinical contexts, continue to perpetuate weight stigma.

Finally, it is important to reflect on the impact that the COVID-19 pandemic had on our study and participants. We conducted our research in 2020, at a time when there was collective anxiety about weight gain during lockdown and extensive media messages about higher body weights being a risk factor for COVID-19 complications. It is likely that this historic moment reinforced already existing weight stigma [68–70].

Reflecting on the authors' motivations and positionality with regard to this research, we recognize our privileges as white, middle/upper-class researchers. Also, as of this writing, none of us have a higher body weight. Despite this limitation, we all have been extensively studying weight stigma and the critical weight literature for years. Also, in our previous research, we

have been actively listening to women with a higher body weight for years–in some cases, for decades.

Study limitations include the sometimes lengthy periods of elapsed time between the surgeries of participants and the interviews we conducted with them, which may have inconsistently influenced responses. This differential time lapse between the surgeries and participation in the study is, however, similar to other studies with post-bariatric population, which included participants with a mean of 7.7- and 13.7-years post-surgery [71]. Furthermore, the varied lengths of time since surgery meant we captured a diverse set of experiences post-surgery, as the women participants developed increasingly divergent trajectories after the first post-operative months.

## Conclusion

Our Brazilian participants faced stigma related to having bariatric surgery, as well as to their body weight. Many of the themes (e.g., surgery as "cheating" and the need to constantly perform self-monitoring) are similar to those identified in studies in the U.S. and Europe. An age-based thematic description of the views and experiences of Brazilian women who underwent bariatric surgery illuminates, in particular, that older women's experiences were more negative than those documented in the weight stigma literature to date. Our results show that people demanded linear weight loss post-surgery but also connected a certain appearance of thinness to "ugliness", "sickness", and, among the older women, "aging". Thus, a complex web of factors determined which bodies were considered acceptable or not across different lifecycles. The older women also frequently heard from the people around them that the side effects of the surgery were worse than having a higher body weight. All these aspects point to a complicated, possibly globalized or globalizing [72] connection between weight and age stigmas. We suggest future studies in the Global South focus on this understudied phenomenon. Also, future research should explore whether a targeted approach might be more effective, for example, an approach that would emphasize the importance of developing coping strategies with respect to experiences of stigma and discrimination after surgery.

## Supporting information

**S1 File. Script of the semi-structured interviews.** In this supporting material we present the script that guided the semi-structured interviews.
(DOCX)

**S2 File. Schedule of the individual semi-structured interviews.** In this supporting material we present the schedule of the individual semi-structured interviews.
(XLSX)

## Author Contributions

**Conceptualization:** Ramiro Fernandez Unsain, Fernanda Baeza Scagliusi.

**Data curation:** Mariana Dimitrov Ulian.

**Formal analysis:** Mariana Dimitrov Ulian, Ramiro Fernandez Unsain.

**Funding acquisition:** Fernanda Baeza Scagliusi.

**Investigation:** Mariana Dimitrov Ulian.

**Methodology:** Mariana Dimitrov Ulian, Ramiro Fernandez Unsain.

**Project administration:** Mariana Dimitrov Ulian.

**Resources:** Ruth Rocha Franco, Marco Aurélio Santo, Bruno Gualano.

**Supervision:** Fernanda Baeza Scagliusi.

**Writing – original draft:** Mariana Dimitrov Ulian.

**Writing – review & editing:** Ramiro Fernandez Unsain, Ruth Rocha Franco, Marco Aurélio Santo, Alexandra Brewis, Sarah Trainer, Cindi SturtzSreetharan, Amber Wutich, Bruno Gualano, Fernanda Baeza Scagliusi.

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
