## [Decision Letter · Decision Letter 0]

28 Dec 2022

PONE-D-22-24534If the weight is gone, is the stigma also gone? A qualitative study of the perceptions and experiences of Brazilian older and younger women who underwent bariatric surgeryPLOS ONE

Dear Dr. Dimitrov Ulian,

Thank you for submitting your manuscript to PLOS ONE. After careful consideration, we feel that it has merit but does not fully meet PLOS ONE’s publication criteria as it currently stands. Therefore, we invite you to submit a revised version of the manuscript that addresses the points raised during the review process.

We look forward to receiving your revised manuscript.

Kind regards,

Carlos Magno Castelo Branco Fortaleza, M.D., Ph.D.

Academic Editor

PLOS ONE

Journal Requirements:

2. Please provide additional details regarding participant consent. In the ethics statement in the Methods and online submission information, please ensure that you have specified whether: 1) whether the ethics committee approved the verbal/oral consent procedure, 2) why written consent could not be obtained, and 3) how verbal/oral consent was recorded. If your study included minors, please state whether you obtained consent from parents or guardians in these cases. If the need for consent was waived by the ethics committee, please include this information.

Furthermore, please clarify how digital consent was obtained and documented

Additional Editor Comments:

The manuscript focuses on a relevant area for research and their results are relevant. There are, however, some flaws that must be corrected. Special attention must be directed to Reviewer #2 comments on the use of stigmatizing language and the "visual analysis" mentioned in lines #208-2015. Methodology section should be improved, including Reviewer #1 recommendation of better description of the characteristics of healthcare and follow up of study participants.The text also requires extensive rewriting to achieve appropriate standards of readability.

Reviewers' comments:

Reviewer's Responses to Questions

**Comments to the Author**

1. Is the manuscript technically sound, and do the data support the conclusions?

Reviewer #1: Yes

Reviewer #2: Partly

2. Has the statistical analysis been performed appropriately and rigorously? 

Reviewer #1: N/A

Reviewer #2: N/A

3. Have the authors made all data underlying the findings in their manuscript fully available?

Reviewer #1: No

Reviewer #2: No

4. Is the manuscript presented in an intelligible fashion and written in standard English?

Reviewer #1: Yes

Reviewer #2: No

5. Review Comments to the Author

Reviewer #1: Introduction

Line 72-72 – “In Brazil, bariatric procedures have been available at no cost to patients since 1999 73 through the Unified Public Health System or the Sistema Único de Saúde [10]” but even so, it is necessary to consider how is the access to this intervention, as the demand for the procedure is certainly very high. Clarify regional differences and the issue of equity.

Methods - Research Setting and Study Population

The autors inform that patients being followed by the Bariatric Surgery Outpatient Clinics of HCFMUSP.

Describe better how women get there, referral criteria and how they are followed up, how often and for how long is the follow-up, before/after surgery, and by which professionals.

In the lines 650-651 the autors indicate “Although our participants also had pre-operative support groups, this support may have been insufficient to combat stressful social messaging.” This suggests that there is no further follow-up, but the women were recruited at this service after the surgery.

The script used for the interviews was not presented, which guiding questions and the previous experience of the researcher (Author1) was not clarified for carrying out both the interviews and data analysis - this aspect is very relevant for the quality of qualitative research.

Results

On Table 1, complete your title indicating location/year. And explain how this information on the women's characteristics was obtained, whether directly with the women or clinic provided, from a medical record.

Reviewer #2: The links between weight and stigma are a growing research area, and it is encouraging to see more research like this in the Global South specifically. Currently, however, the manuscript requires major revisions to most sections. I’ve provided comments below to try and guide where these revisions would be beneficial.

Title

The title would stand alone better, be more grammatically correct, and be clearer without the question. For example, as “A qualitative study of Brazilian women’s’ perceptions and experiences of weight stigma after bariatric surgery”. I echo the same point for the short title. Perhaps a reframe to “A qualitative study of stigma after bariatric surgery”.

Abstract

• Can you please reframe the following into a research aim/objective (rather than a question): “However, how do bariatric patients perceive and internalize this stigma in different life stages? Specifically, is older age a risk factor?”. The current framing is a little too colloquial.

• Can you explicitly state the method of data analysis used? E.g., was it thematic analysis or something different?

• The abstract is missing a conclusion. Can you please add a sentence concluding your findings, and perhaps suggesting how the findings may be used in the future? You could make your results more concise if you need more word count to allow this.

General

• Can you explain if and how you included patient and public involvement in this work? For example, what stages were they involved and what activities/input did they have.

• Throughout the manuscript, there are quantifying words or sentences for example: ‘For all participants’ or ‘Some of the women’. Please delete the quantifying words.

• The manuscript would benefit greatly from a review of the grammar/scientific writing.

Introduction

• Can the tone of the following sentence be softened: “all of which induce severe weight loss through the surgical reduction of the stomach and/or intestines”. For example, to “all of which aim to induce severe weight loss through the surgical reduction of the stomach and/or intestines”

• Please add references to support: “Bariatric patients often present for surgery with a history of complex anxieties and negative social experiences stemming from weight-related stigma.”

• The word ‘deviant’ isn’t necessary to the message in the sentence on line 82 in the second paragraph of the introduction – can you please remove.

• In the definition of weight stigma by Brewis, stigmatising language itself is used (“being overweight or obese”). Can you please reframe this to move away from this language and use person-first language instead?

• In the introduction (and manuscript) you move between using the term weight stigma and weight-related stigma. Can you please amend the manuscript so you use consistent language throughout?

• In the introduction, you use the stigmatising language of “fat”. Can you please address this throughout the manuscript to use person-first language?

• The introduction is very lengthy and consistently takes multiple paragraphs to make a point that could be made more clearly in one paragraph. Could you please review the introduction to see where content could be made more concise, and therefore clearer for the reader to follow?

• Can you soften the tone in the following sentence: “Weight loss induced by surgical intervention is so rapid that it produces a great deal of excess skin and it also begins at the “ends,” i.e., people tend to lose weight in their heads, arms, and legs before they lose it in their torsos”. For example, to: “Weight loss induced by surgical intervention is so rapid that it often produces a great deal of excess skin. Furthermore, people tend to lose weight in their heads, arms, and legs before they lose it in their torsos, influencing their body shape and size”.

• Despite the length of the introduction, it does not note the potential implications of addressing the research aim. For example, how will this inform public health?

Methods

• Can you please move the ethical approval statement to the start of the methods section?

• Overall, there is a great lack of detail in the data analysis section. Please review and add more detail here to aid the reader to thoroughly understand what you have done with enough detail for them to replicate the processes. For example: You note participants who fit the ‘proposed profile’ were invited to interview - what is the proposed profile? What sampling method did you use? Did you use purposeful sampling, snowball, opportunist? How were the interviews transcribed? Was this by an external company or by the lead author?

• You note in the data collection section that “This two-encounters approach proved to be quite effective in obtaining data and resulted in the chance to complete the interview protocol in a way that was practical and feasible for the participants’ schedules”. I’m unclear how conducting two interviews would be more practical and feasible for someone’s schedule than conducting one interview – can you explain? Also, “proved to be quite effective in obtaining data” is very unclear and non-specific. I’m not sure the value this sentence brings or the message you are trying to communicate.

• You state that “Notably, the time gap between the first and second interviews led to increased reflexivity among the participants and the researcher, who reviewed the recording of the first encounter to note gaps that needed to be covered in the second.” Reflexivity is the acknowledgement of your role and influence on the research process, so it does not make sense in this sentence. Perhaps you were meaning data quality? I’d also suggest softening the tone of many of these sentences. You cannot be certain that the time gap you had led to increased (presumably) data quality, but you could say that was the aim/purpose of the time gap.

• I strongly oppose to many of the messages from line 208 to 215, and feel the messages themselves exacerbate the stigmatisation of those living with overweight or obesity. If you are analysing participants experiences (e.g., feeling like they had not lost enough weight), then it is not your role and it is inappropriate to say that you must visually assess/observe their bodies to validate this. Furthermore, the notation of having to “rely on her description” is highly problematic. If you are analysing someone’s experiences, then the analysis and resultant publication should be aiming to represent the voices of the participant, not your assessment as to whether these experiences are reliable. Please remove this paragraph.

• Lines 227 to 229 are hard to follow – can you please rephrase?

• In lines 228-229, you note themes as emerging as significant. These is vast amounts of literature noting themes as “emergent” to be problematic as it suggests that the themes always existed within the data, and voids the influence of our positionality as the researcher on the process. Can you please review this throughout the manuscript to remove this concept?

• In lines 231-233, you state that you identified excerpts of interviews relevant to the themes of interest, but you have not explained the process of theme development at this stage. How did you develop the themes of interest?

• You have not described how the interview schedule was produced or referenced to a copy of the interview schedule in the supplements. Please add these details.

• There lacks a section on the researcher positionality and how they managed the impact of this through reflexivity. Please can the authors address this.

Results

• In Table 1, please remove the percent and standard deviation. These are not generally used in qualitative studies.

• You presented quotations in multiple languages, but have not noted this translation process in the methods section. Please add this information.

• Throughout the results section, you state findings in affirmative language – for example, “she asked if this man had…”. A better/more accurate way of describing the findings is to include acknowledgement of the interpretation of their accounts by including prefaces – for example, such as “the participant described/expressed that she asked the man...”. It would strengthen the manuscript to review and revise these areas.

• The final lines of the first theme (lines 314 to 317) don’t seem to fit with the narrative of the rest of the theme. Can you please review and either remove or clarify their fit with the narrative?

• Line 323 – are these self-appointed by the participant or by their peers?

• Between lines 329 to 335, you describe others having concerns over the participants health. Could you please include a supporting quotation here?

• On lines 348 to 350, you introduce new literature within the results section. You shouldn’t incorporate other literature in this section so could you remove and rephrase this section. For example, you could simply say “Participants who described experiencing small weight regain after weight loss described being reprimanded by others”.

• Line 360 – is “melt off” a quote from a participant or your own writing? If a quote, can quotations marks please be added. If it is your own writing, can it be rephrased to be more in line with the language and writing style of the piece.

• Again in lines 366 to 370, you introduce new literature and compare findings to existing evidence. Please remove this from the results. This is repeated in areas throughout the results section – can you please review this throughout.

• Lines 391 to 392 begin to discuss the results, rather than present the results. Can you please move this to the discussion or remove?

• Line 393 “aggressive” and line 396 “mad”– were these words used by the participants or is this your own language choice?

• Line 400 to 401 – did participants say that the advice was not backed up by evidence, or is this your own commentary? If the latter, please move to the discussion or remove.

• Line 423 – did the participant describe the foods as “fattening foods” themselves? If so, can you please add quotation marks. If not, can you please remove/reframe this concept.

• Lines 444 to 450 do not belong in the results section altogether, and the key points should only remain in the introduction. I recommend to remove altogether and keep the focus to the findings of this particular study.

• Lines 470 to 472 – I recommend to rephrase these lines. Currently, this reads as your perception rather than a reflection of participants voices and experiences.

• Unit of meaning 4 – this title includes non-person first language. Can this please be amended.

• Unit of meaning 4 does not appear to have a well-bounded message or narrative. The points within feel quite distinct from one another at times, and they do not always relate to the overarching title given to the theme. I think this theme needs a reworking entirely to reclarify what the key messages are, and refocused the text to make those clearer.

• Overall, the themes are extremely long which makes it difficult at times to follow the key points – they can get lost in the length of text. Can you review the themes and identify where things could be cut down or made more concise? For example, when a message is communicated, you often include many examples and quotations, and this many are not always necessary to support and evidence the message communicated. You could look to cut down these areas. In addition, much of the themes feels like a reproduction of the contributing interviews, rather than a summative/themed narrative cutting across the interviews, thus contributing to the length of the themes and the dilution of the messages within.

• A key aim of the paper was to compare the differences in experiences of younger and older women, but from how the results are presented (i.e., experiences of each group intertwined throughout, without much comparative analytical commentary) it is not possible to do this. Have you considered separately the results in each theme by younger and older to make clearer distinctions ofaz these experiences?

Discussion

• Lines 561-562. In the introduction, the aim is framed to compare younger and older women, rather than to focused explicitly on the experiences of older women as it is framed in this line. Can you please amend this line to be consistent?

• Paragraph 2 (lines 576 to 589) mostly represents the findings of the study, and there is limited comparison to other available literature. It is also a little to strong to claim this finding to be ‘novel’ due to perceived strength of messaging, and I recommend softening the tone here. In addition, the end of the paragraph lacks cited literature to support the statements made (i.e., lines 586-589). The language/grammar would benefit from reviewing in this area too.

• Consistently throughout the discussion, the paragraphs tend to reproduce the findings of the study, with very limited contextualisation or comparison with the wider literature base.

• Lines 620 to 624 – this sentence is extremely long and hard to follow, and it also introduces several new, large points (e.g., Foucault’s notion of biopower). These concepts should be introduced in the introduction if they are to be discussed because, as currently written, this section does not give enough context or background information for the reader to fully conceptualise the messages intended.

• Paragraph on lines 620 to 634 does not seem to fit with the narrative of the entire manuscript, and the messages made feel extremely distinct and off-topic. I’m unclear how your findings compare to this and fit within these messages. In addition, in areas, it reads to have been influenced by opinions rather than a balanced discussion of this complex area. For example, the description of weight management to be “punitive” and “authoritarian” and aligned with assumptions that “those with a higher body weight lack knowledge and understanding about health practices, make poor choices, and refuse to take responsibility for their health and well-being”. If this section is to remain, it would require reworking to provide a scientifically critical and balanced discussion from multiple lens.

• Line 631 – we do not tend to gender the authors of literature we are citing as we cannot assume one’s gender. Can you please review this throughout the manuscript.

• The manuscript lacks a section highlighting the limitations of the study. Please review and add this to the manuscript. This should include (among others) – the impact of researcher positionality and the recall bias due to the long time span between the surgeries of participants.

Conclusion

• Lines 679-682, and lines 686-687. Insufficient comparison between the groups has been presented through the manuscript as currently written to make a clear conclusion about the differences in experiences. I would recommend reworking the results section to make this potentially possible to conclude, or remove these conclusions.

6. PLOS authors have the option to publish the peer review history of their article (what does this mean?). If published, this will include your full peer review and any attached files.

Reviewer #1: No

Reviewer #2: No

---

## [Author Response · Author response to Decision Letter 0]

9 Feb 2023

Dear Editor Professor Dra. Emily Chenette:

We are pleased to re-submit the revised version of the manuscript “If the weight is gone, is the stigma also gone? A qualitative study of the perceptions and experiences of Brazilian older and younger women who underwent bariatric surgery”. We highlight that after the reviewers’ comments, the title has changed to “A qualitative study of Brazilian women’s perceptions and experiences of weight stigma after bariatric surgery”, as suggested. We have extensively revised the manuscript according to the suggestions made by the referee and the editor. An item-by-item response is presented below. All changes in the manuscript are highlighted with tracked changes. We hope that these changes will meet with your approval. 

- Editor:

Article formatting

Editor comment: Please ensure that your manuscript meets PLOS ONE’s style requirements, including those for file naming. The PLOS ONE style templates can be found at https://journals.plos.org/plosone/s/file?id=wjVg/PLOSOne_formatting_sample_main_body.pdf and 

Response: We have formatted the manuscript accordingly.

Methods

Editor comment: Please provide additional details regarding participant consent. In the ethics statement in the Methods and online submission information, please ensure that you have specified whether: 1) whether the ethics committee approved the verbal/oral consent procedure, 2) why written consent could not be obtained, and 3) how verbal/oral consent was recorded. If your study included minors, please state whether you obtained consent from parents or guardians in these cases. If the need for consent was waived by the ethics committee, please include this information.

Response: We have clarified these aspects in the Methods section and in the online submission information. Regarding the question about why written consent could not be obtained, we conducted the research during the pandemic, and therefore, the interviews were conducted remotely via WhatsApp video calls. Asking the participants for a written consent could be demanding for them, and possibly, excluding. For example, as they would have to print, then scan the signed document and finally send to Author 1, some of them, especially the older women, could have difficulty to do so. Thus, we understood that both oral consent and digitally recorded informed consent would provide access to all participants.

Data availability

Editor comment: In your Data Availability statement, you have not specified where the minimal data set underlying the results described in your manuscript can be found. PLOS defines a study’s minimal data set as the underlying data used to reach the conclusions drawn in the manuscript and any additional data required to replicate the reported study findings in their entirety. All PLOS journals require that the minimal data set be made fully available. For more information about our data policy, please see http://journals.plos.org/plosone/s/data-availability.

“Upon re-submitting your revised manuscript, please upload your study’s minimal underlying data set as either Supporting Information files or to a stable, public repository and include the relevant URLs, DOIs, or accession numbers within your revised cover letter. For a list of acceptable repositories, please see http://journals.plos.org/plosone/s/data-availability#loc-recommended-repositories. Any potentially identifying patient information must be fully anonymized.

Response: We have included or data set, which is available in a public repository from the University of São Paulo, available at: https://repositorio.uspdigital.usp.br/handle/item/399

General comments

Editor comment: The manuscript focuses on a relevant area for research and their results are relevant. There are, however, some flaws that must be corrected. Special attention must be directed to Reviewer #2 comments on the use of stigmatizing language and the “visual analysis” mentioned in lines #208-2015. Methodology section should be improved, including Reviewer #1 recommendation of better description of the characteristics of healthcare and follow up of study participants. The text also requires extensive rewriting to achieve appropriate standards of readability.

Response: We have addressed the aspects mentioned and they are fully addressed in the responses below. 

- Reviewer’s Responses to Questions

Comments to the Author

Reviewers’ comments: 1. Is the manuscript technically sound, and do the data support the conclusions?

Reviewer #1: Yes

Reviewer #2: Partly

Response: We have extensively reviewed the manuscript to address these matters.

2. Has the statistical analysis been performed appropriately and rigorously?

Reviewer #1: N/A

Reviewer #2: N/A

3. Have the authors made all data underlying the findings in their manuscript fully available?

Reviewer #1: No

Reviewer #2: No

Response: We have included or data set, which is available in a public repository from the University of São Paulo, available at: https://repositorio.uspdigital.usp.br/handle/item/399

4. Is the manuscript presented in an intelligible fashion and written in standard English?

Reviewer #1: Yes

Reviewer #2: No

Response: The manuscript was submitted to an extensive English revision by native English speakers.

- Reviewer 1

Introduction

Reviewer comment: Line 72-72 – “In Brazil, bariatric procedures have been available at no cost to patients since 1999 73 through the Unified Public Health System or the Sistema Único de Saúde [10]” but even so, it is necessary to consider how is the access to this intervention, as the demand for the procedure is certainly very high. Clarify regional differences and the issue of equity.

Response: We have addressed these aspects in the Introduction section.

Methods

Reviewer comment: The authors inform that patients being followed by the Bariatric Surgery Outpatient Clinics of HCFMUSP. Describe better how women get there, referral criteria and how they are followed up, how often and for how long is the follow-up, before/after surgery, and by which professionals.

Response: We have these aspects in the Methods section.

Methods

Reviewer comment: The script used for the interviews was not presented, which guiding questions and the previous experience of the researcher (Author1) was not clarified for carrying out both the interviews and data analysis - this aspect is very relevant for the quality of qualitative research.

Response: In the reviewed version of the manuscript, we included the script that guided the interviews (Supporting Information 1). Also, in the Methods section, we included the previous experience and training of Author 1 regarding conducted individual interviews.

Results

Reviewer comment: On Table 1, complete your title indicating location/year. And explain how this information on the women’s characteristics was obtained, whether directly with the women or clinic provided, from a medical record.

Response: We completed the title indicating location and year. The information on the women’s characteristics was obtained directly with the participants, during the individual interviews. We added this information in the Results section. These questions can be seen in the script that guided the interviews, submitted as a Supplementary Material. 

Discussion

Reviewer comment: In the lines 650-651 the authors indicate “Although our participants also had pre-operative support groups, this support may have been insufficient to combat stressful social messaging.” This suggests that there is no further follow-up, but the women were recruited at this service after the surgery.

Response: We have clarified this aspect in the Discussion section.

- Reviewer 2

General

Reviewer comment: Can you explain if and how you included patient and public involvement in this work? For example, what stages were they involved and what activities/input did they have.

Response: Participants were patients who underwent bariatric surgery at Hospital das Clínicas of the School of Medicine, University of São Paulo. Their involvement with the research included the individual semi-structured interviews. They were contacted by Author 1 and, if they agreed to participate, the interview was scheduled. These procedures are extensively described in the Methods section.

General

Reviewer comment: Throughout the manuscript, there are quantifying words or sentences for example: ‘For all participants’ or ‘Some of the women’. Please delete the quantifying words.

Response: These changes were made accordingly. 

General

Reviewer comment: The manuscript would benefit greatly from a review of the grammar/scientific writing.

Response: The manuscript was submitted to an extensive English revision by native English speakers.

Title

Reviewer comment: The title would stand alone better, be more grammatically correct, and be clearer without the question. For example, as “A qualitative study of Brazilian women’s’ perceptions and experiences of weight stigma after bariatric surgery”. I echo the same point for the short title. Perhaps a reframe to “A qualitative study of stigma after bariatric surgery”.

Response: The changes in the Title were made accordingly. 

Abstract

Reviewer comment: Can you please reframe the following into a research aim/objective (rather than a question): “However, how do bariatric patients perceive and internalize this stigma in different life stages? Specifically, is older age a risk factor?”. The current framing is a little too colloquial. Can you explicitly state the method of data analysis used? E.g., was it thematic analysis or something different? The abstract is missing a conclusion. Can you please add a sentence concluding your findings, and perhaps suggesting how the findings may be used in the future? You could make your results more concise if you need more word count to allow this.

Response: The changes in the Abstract were made accordingly. 

Introduction

Reviewer comment: Can the tone of the following sentence be softened: “all of which induce severe weight loss through the surgical reduction of the stomach and/or intestines”. For example, to “all of which aim to induce severe weight loss through the surgical reduction of the stomach and/or intestines”

Response: The change was made accordingly. 

Introduction

Reviewer comment: Please add references to support: “Bariatric patients often present for surgery with a history of complex anxieties and negative social experiences stemming from weight-related stigma.”

Response: We added the following references: 1) da Silva SSP and da Costa Maia Â (2012) Obesity and treatment meanings in bariatric surgery candidates: a qualitative study. Obesity surgery 22(11), 1714-1722. https://doi.org/10.1007/s11695-012-0716-y; 2) Trainer S, Brewis, A and Wutich A (2017) Not ‘taking the easy way out’: reframing bariatric surgery from low-effort weight loss to hard work. Anthropology & Medicine 24(1), 96-110. https://doi.org/10.1080/13648470.2016.1249339; 3) Hansen, B and Dye, MH (2018) Damned if You Do, Damned if You Don’t: The Stigma of Weight Loss Surgery, Deviant Behavior, 39:2, 137-147, DOI: 10.1080/01639625.2016.1263081

Introduction

Reviewer comment: The word ‘deviant’ isn’t necessary to the message in the sentence on line 82 in the second paragraph of the introduction – can you please remove.

Response: The change was made accordingly. 

Introduction

Reviewer comment: In the definition of weight stigma by Brewis, stigmatising language itself is used (“being overweight or obese”). Can you please reframe this to move away from this language and use person-first language instead?

Response: This was a direct citation of the term that the Brewis used. Nonetheless, we understand the need to change, and we have done such change as the following: “Following Brewis [16] we define weight stigma as moral discrediting that people experience from others or apply to themselves because of the negative social meanings attached to the size of bodies”. 

Introduction

Reviewer comment: In the introduction (and manuscript) you move between using the term weight stigma and weight-related stigma. Can you please amend the manuscript so you use consistent language throughout?

Response: The change was made accordingly and used the term “weight stigma” throughout the manuscript. 

Introduction

Reviewer comment: In the introduction, you use the stigmatising language of “fat”. Can you please address this throughout the manuscript to use person-first language?

Response: The change was made accordingly. We just kept the word “fat” throughout the manuscript when it was an emic word or when referring to authors that employed this word. 

Introduction

Reviewer comment: The introduction is very lengthy and consistently takes multiple paragraphs to make a point that could be made more clearly in one paragraph. Could you please review the introduction to see where content could be made more concise, and therefore clearer for the reader to follow?

Response: The change was made accordingly.

Introduction

Reviewer comment: Can you soften the tone in the following sentence: “Weight loss induced by surgical intervention is so rapid that it produces a great deal of excess skin and it also begins at the “ends,” i.e., people tend to lose weight in their heads, arms, and legs before they lose it in their torsos”. For example, to: “Weight loss induced by surgical intervention is so rapid that it often produces a great deal of excess skin. Furthermore, people tend to lose weight in their heads, arms, and legs before they lose it in their torsos, influencing their body shape and size”.

Response: The change was made accordingly.

Introduction

Reviewer comment: Despite the length of the introduction, it does not note the potential implications of addressing the research aim. For example, how will this inform public health?

Response: The change was made accordingly.

Methods

Reviewer comment: Can you please move the ethical approval statement to the start of the methods section?

Response: The change was made accordingly.

Methods

Reviewer comment: Overall, there is a great lack of detail in the data analysis section. Please review and add more detail here to aid the reader to thoroughly understand what you have done with enough detail for them to replicate the processes. For example: You note participants who fit the ‘proposed profile’ were invited to interview - what is the proposed profile? What sampling method did you use? Did you use purposeful sampling, snowball, opportunist? How were the interviews transcribed? Was this by an external company or by the lead author?

Response: The change was made accordingly.

Methods

Reviewer comment: You note in the data collection section that “This two-encounters approach proved to be quite effective in obtaining data and resulted in the chance to complete the interview protocol in a way that was practical and feasible for the participants’ schedules”. I’m unclear how conducting two interviews would be more practical and feasible for someone’s schedule than conducting one interview – can you explain? Also, “proved to be quite effective in obtaining data” is very unclear and non-specific. I’m not sure the value this sentence brings or the message you are trying to communicate.

Response: Our interviews were long, lasting an average of three hours or more. When engaging with the participants, Author 1 exposed the estimate time of the interview and the participants asked to have the interview divided in two encounters, and sometimes, a third encounter was necessary. This was because it was much more feasible to them to provide Author 1 two (or three) shorter encounters than one longer single encounter. In that sense, if we insisted on having the interview in one single encounter, most of the participants could feel discouraged to participate. Therefore, this is what we meant by this proving “to be quite effective in obtaining data”.

Methods

Reviewer comment: You state that “Notably, the time gap between the first and second interviews led to increased reflexivity among the participants and the researcher, who reviewed the recording of the first encounter to note gaps that needed to be covered in the second.” Reflexivity is the acknowledgement of your role and influence on the research process, so it does not make sense in this sentence. Perhaps you were meaning data quality? I’d also suggest softening the tone of many of these sentences. You cannot be certain that the time gap you had led to increased (presumably) data quality, but you could say that was the aim/purpose of the time gap.

Response: The change was made accordingly.

Methods

Reviewer comment: I strongly oppose to many of the messages from line 208 to 215, and feel the messages themselves exacerbate the stigmatisation of those living with overweight or obesity. If you are analysing participants experiences (e.g., feeling like they had not lost enough weight), then it is not your role and it is inappropriate to say that you must visually assess/observe their bodies to validate this. Furthermore, the notation of having to “rely on her description” is highly problematic. If you are analysing someone’s experiences, then the analysis and resultant publication should be aiming to represent the voices of the participant, not your assessment as to whether these experiences are reliable. Please remove this paragraph.

Response: We removed this paragraph. 

Methods

Reviewer comment: Lines 227 to 229 are hard to follow – can you please rephrase?

Response: We removed this paragraph. 

Methods

Reviewer comment: In lines 228-229, you note themes as emerging as significant. These is vast amounts of literature noting themes as “emergent” to be problematic as it suggests that the themes always existed within the data, and voids the influence of our positionality as the researcher on the process. Can you please review this throughout the manuscript to remove this concept?

Response: We reviewed the manuscript and removed this concept as requested. Also, we clarified about the development of themes in the Data Analysis subsection. 

Methods

Reviewer comment: In lines 231-233, you state that you identified excerpts of interviews relevant to the themes of interest, but you have not explained the process of theme development at this stage. How did you develop the themes of interest?

Response: We have clarified the data analysis, as well as the development of themes. It is in the Data Analysis subsection.

Methods

Reviewer comment: You have not described how the interview schedule was produced or referenced to a copy of the interview schedule in the supplements. Please add these details.

Response: We have included a copy of the interview schedule in the Supporting Information. We clarify that we are presenting both the interview script and schedule as Supporting Informations.

Methods

Reviewer comment: There lacks a section on the researcher positionality and how they managed the impact of this through reflexivity. Please can the authors address this.

Response: We have addressed these aspects accordingly at the end of Discussion section. 

Results

Reviewer comment: In Table 1, please remove the percent and standard deviation. These are not generally used in qualitative studies. 

Response: We removed the percent and standard deviation from Table 1 accordingly.

Results

Reviewer comment: You presented quotations in multiple languages, but have not noted this translation process in the methods section. Please add this information.

Response: We have added this information accordingly. 

Results

Reviewer comment: Throughout the results section, you state findings in affirmative language – for example, “she asked if this man had…”. A better/more accurate way of describing the findings is to include acknowledgement of the interpretation of their accounts by including prefaces – for example, such as “the participant described/expressed that she asked the man...”. It would strengthen the manuscript to review and revise these areas.

Response: We have reviewed and revised these areas accordingly. 

Results

Reviewer comment: The final lines of the first theme (lines 314 to 317) don’t seem to fit with the narrative of the rest of the theme. Can you please review and either remove or clarify their fit with the narrative?

Response: We have removed these lines.

Results

Reviewer comment: Line 323 – are these self-appointed by the participant or by their peers?

Response: The comments were expressed by other people. We changed the text to clarify the message. 

Results

Reviewer comment: Between lines 329 to 335, you describe others having concerns over the participants health. Could you please include a supporting quotation here?

Response: We included supporting quotations in this paragraph. 

Results

Reviewer comment: On lines 348 to 350, you introduce new literature within the results section. You shouldn’t incorporate other literature in this section so could you remove and rephrase this section. For example, you could simply say “Participants who described experiencing small weight regain after weight loss described being reprimanded by others”.

Response: We removed citation and made the change in the paragraph as suggested. 

Results

Reviewer comment: Line 360 – is “melt off” a quote from a participant or your own writing? If a quote, can quotations marks please be added. If it is your own writing, can it be rephrased to be more in line with the language and writing style of the piece.

Response: The use of the word “melt off” was a choice of the Authors. Nonetheless, we agree that it was not in line with the language and writing style of the manuscript and have changed it accordingly.

Results

Reviewer comment: Again in lines 366 to 370, you introduce new literature and compare findings to existing evidence. Please remove this from the results. This is repeated in areas throughout the results section – can you please review this throughout.

Response: We removed lines 366 to 370 from the results and revised the entire section to remove other citations that were previously included. 

Results

Reviewer comment: Lines 391 to 392 begin to discuss the results, rather than present the results. Can you please move this to the discussion or remove?

Response: We removed lines 391 to 392 as requested. 

Results

Reviewer comment: Line 393 “aggressive” and line 396 “mad”– were these words used by the participants or is this your own language choice?

Response: We changed these lines to clarify that these words were used by the participants. 

Results

Reviewer comment: Line 400 to 401 – did participants say that the advice was not backed up by evidence, or is this your own commentary? If the latter, please move to the discussion or remove.

Response: We removed lines 400 to 401 as requested. 

Results

Reviewer comment: Line 423 – did the participant describe the foods as “fattening foods” themselves? If so, can you please add quotation marks. If not, can you please remove/reframe this concept.

Response: We have clarified line 423 to show that this was how the participants described the foods and included a supporting quotation.

Results

Reviewer comment: Lines 444 to 450 do not belong in the results section altogether, and the key points should only remain in the introduction. I recommend to remove altogether and keep the focus to the findings of this particular study.

Response: We removed lines 444 to 450 as requested. 

Results

Reviewer comment: Lines 470 to 472 – I recommend to rephrase these lines. Currently, this reads as your perception rather than a reflection of participants voices and experiences.

Response: We have removed lines 470 to 472 as requested.

Results

Reviewer comment: Unit of meaning 4 – this title includes non-person first language. Can this please be amended.

Response: We have included person-first language, except when the participant herself voiced a non-person first language, such as “fat” (in Brazilian Portuguese, “gorda”).

Results

Reviewer comment: Unit of meaning 4 does not appear to have a well-bounded message or narrative. The points within feel quite distinct from one another at times, and they do not always relate to the overarching title given to the theme. I think this theme needs a reworking entirely to reclarify what the key messages are, and refocused the text to make those clearer.

Response: We have reworked on this theme and changed its title to clarify the message. 

Results

Reviewer comment: Overall, the themes are extremely long which makes it difficult at times to follow the key points – they can get lost in the length of text. Can you review the themes and identify where things could be cut down or made more concise? For example, when a message is communicated, you often include many examples and quotations, and this many are not always necessary to support and evidence the message communicated. You could look to cut down these areas. In addition, much of the themes feels like a reproduction of the contributing interviews, rather than a summative/themed narrative cutting across the interviews, thus contributing to the length of the themes and the dilution of the messages within.

Response: We have extensively revised the Results section, cut down some information and made it more concise. 

Results

Reviewer comment: A key aim of the paper was to compare the differences in experiences of younger and older women, but from how the results are presented (i.e., experiences of each group intertwined throughout, without much comparative analytical commentary) it is not possible to do this. Have you considered separately the results in each theme by younger and older to make clearer distinctions of these experiences?

Response: As suggested, we have separated the results in each theme by: 1) Common experiences among the younger and older participants and; 2) The experiences of the older participants.

Discussion

Reviewer comment: Lines 561-562. In the introduction, the aim is framed to compare younger and older women, rather than to focused explicitly on the experiences of older women as it is framed in this line. Can you please amend this line to be consistent?

Response: The change was made accordingly.

Discussion

Reviewer comment: Paragraph 2 (lines 576 to 589) mostly represents the findings of the study, and there is limited comparison to other available literature. It is also a little to strong to claim this finding to be ‘novel’ due to perceived strength of messaging, and I recommend softening the tone here. In addition, the end of the paragraph lacks cited literature to support the statements made (i.e., lines 586-589). The language/grammar would benefit from reviewing in this area too.

Response: We have expanded the comparison to other available literature in Paragraph 2 and softened the tone regarding the novel finding. Finally, the manuscript was submitted to an extensive English revision by native English speakers.

Discussion

Reviewer comment: Consistently throughout the discussion, the paragraphs tend to reproduce the findings of the study, with very limited contextualisation or comparison with the wider literature base.

Response: We revisited the Discussion and made extensive changes, as well as included more contextualization or comparison with the wider literature base.

Discussion

Reviewer comment: Lines 620 to 624 – this sentence is extremely long and hard to follow, and it also introduces several new, large points (e.g., Foucault’s notion of biopower). These concepts should be introduced in the introduction if they are to be discussed because, as currently written, this section does not give enough context or background information for the reader to fully conceptualise the messages intended.

Response: To attend the following comment of the Reviewer, we have removed Lines 620 to 634.

Discussion

Reviewer comment: Paragraph on lines 620 to 634 does not seem to fit with the narrative of the entire manuscript, and the messages made feel extremely distinct and off-topic. I’m unclear how your findings compare to this and fit within these messages. In addition, in areas, it reads to have been influenced by opinions rather than a balanced discussion of this complex area. For example, the description of weight management to be “punitive” and “authoritarian” and aligned with assumptions that “those with a higher body weight lack knowledge and understanding about health practices, make poor choices, and refuse to take responsibility for their health and well-being”. If this section is to remain, it would require reworking to provide a scientifically critical and balanced discussion from multiple lens.

Response: We have removed Lines 620 to 634.

Discussion

Reviewer comment: Line 631 – we do not tend to gender the authors of literature we are citing as we cannot assume one’s gender. Can you please review this throughout the manuscript.

Response: We believe it is essential to point out that all the authors who sign this article stand out for investigating the different social markers of difference, including gender and sexuality, from an intersectional perspective. In this sense, the assumption of a particular gender is far from our perspective. If we de-gender an author, we would use the pronouns he and him, which are already gendered. Therefore, we decided to continue to refer to the authors as they refer to themselves as a gender through their productions and first names. We believe that de-gendering authors, without their consent, could become a lack of respect for their gender ascriptions. 

Discussion

Reviewer comment: The manuscript lacks a section highlighting the limitations of the study. Please review and add this to the manuscript. This should include (among others) – the impact of researcher positionality and the recall bias due to the long time span between the surgeries of participants.

Response: We have included a section highlighting the limitations of the study as requested.

Conclusion

Reviewer comment: Lines 679-682, and lines 686-687. Insufficient comparison between the groups has been presented through the manuscript as currently written to make a clear conclusion about the differences in experiences. I would recommend reworking the results section to make this potentially possible to conclude, or remove these conclusions.

Response: We changed the focus of the article to do a description of older and younger women after bariatric surgery. We also have separated the results in each theme by younger and older to make clearer distinctions of these experiences (as highlighted in a response above). With these changes, we believe that the Conclusion is now sufficient support the data presented. 

We thank you again for the valuable contributions of the reviewer and yourself and for the opportunity to submit our manuscript for your consideration.

Kind regards,

The authors.

---

## [Decision Letter · Decision Letter 1]

9 May 2023

PONE-D-22-24534R1A qualitative study of Brazilian women’s perceptions and experiences of weight stigma after bariatric surgeryPLOS ONE

Dear Dr. Dimitrov Ulian,

Thank you for submitting your manuscript to PLOS ONE. After careful consideration, we feel that it has merit but does not fully meet PLOS ONE’s publication criteria as it currently stands. Therefore, we invite you to submit a revised version of the manuscript that addresses the points raised during the review process.

We look forward to receiving your revised manuscript.

Kind regards,

Vidanka Vasilevski

Academic Editor

PLOS ONE

Journal Requirements:

Additional Editor Comments:

The revisions to the study have improved the quality of the paper, however there remain a few minor readibility issues that need to be rectified prior to publication.

Some references need to be added/edited in the introduction section.

The full paper requires English language editing, engaging an editing service may support this. There are many long and complex sentences that need refining also.

In the research setting and study population section, the eligibility criteria includes gendered language (i.e., her). I assume men and women would be eligible for bariatric surgery, therefore referring to 'her' only is not appropriate.

Reviewers' comments:

Reviewer's Responses to Questions

**Comments to the Author**

1. If the authors have adequately addressed your comments raised in a previous round of review and you feel that this manuscript is now acceptable for publication, you may indicate that here to bypass the “Comments to the Author” section, enter your conflict of interest statement in the “Confidential to Editor” section, and submit your "Accept" recommendation.

Reviewer #2: (No Response)

2. Is the manuscript technically sound, and do the data support the conclusions?

Reviewer #2: Yes

3. Has the statistical analysis been performed appropriately and rigorously? 

Reviewer #2: N/A

4. Have the authors made all data underlying the findings in their manuscript fully available?

Reviewer #2: Yes

5. Is the manuscript presented in an intelligible fashion and written in standard English?

Reviewer #2: Yes

6. Review Comments to the Author

Reviewer #2: Thank you for your responses to my comments - the article is reading much better. To note - your responses should direct the reviewer to the page and line numbers where the edits in text have been made and, if possible, provide the edited text within your response also. It makes the process of peer review substantially more time consuming and difficult to not be given direction or detail within responses.

Original reviewer comment: Can you explain if and how you included patient and public involvement in this work? For example, what stages were they involved and what activities/input did they have.

Author response: Participants were patients who underwent bariatric surgery at Hospital das Clínicas of the School of Medicine, University of São Paulo. Their involvement with the research included the individual semi-structured interviews. They were contacted by Author 1 and, if they agreed to participate, the interview was scheduled. These procedures are extensively described in the Methods section.

Reviewer response: Patient and public involvement (PPI) is an active partnership between members of the public and researchers, where the public are involved in aspects of the study/research (such as informing the study design, supporting with analysis). The participants of the study do not count as PPI. You can read more about this on: https://www.rds-sc.nihr.ac.uk/ppi-information-resources/ Can you please provide a response given this explanation of PPI.

Original reviewer comment: Can you please reframe the following into a research aim/objective (rather than a question): “However, how do bariatric patients perceive and internalize this stigma in different life stages? Specifically, is older age a risk factor?”. The current framing is a little too colloquial. Can you explicitly state the method of data analysis used? E.g., was it thematic analysis or something different? The abstract is missing a conclusion. Can you please add a sentence concluding your findings, and perhaps suggesting how the findings may be used in the future? You could make your results more concise if you need more word count to allow this.

Author response: The changes in the Abstract were made accordingly.

Reviewer response: You have not added the method of data analysis used to the abstract – please address this. The conclusion added goes beyond what your findings can support, and should be softened in tone to be more appropriate to the levels of findings. Specifically, the findings do not inform a targeted approach, however they might suggest future research should explore whether a targeted approach might be more effective.

Original reviewer comment: In the introduction, you use the stigmatising language of “fat”. Can you please address this throughout the manuscript to use person-first language?

Author response: The change was made accordingly. We just kept the word “fat throughout the manuscript when it was an emic word or when referring to authors that employed this word.

Reviewer response: Can you please edit the text to have quotation marks (or something similar) to make it clear to the reader than the remaining use of the word ‘fat’ is not your words, but quoting participants etc.

Original reviewer comment: Line 631 – we do not tend to gender the authors of literature we are citing as we cannot assume one’s gender. Can you please review this throughout the manuscript.

Authors response: We believe it is essential to point out that all the authors who sign this article stand out for investigating the different social markers of difference, including gender and sexuality, from an intersectional perspective. In this sense, the assumption of a particular gender is far from our perspective. If we de-gender an author, we would use the pronouns he and him, which are already gendered. Therefore, we decided to continue to refer to the authors as they refer to themselves as a gender through their productions and first names. We believe that de-gendering authors, without their consent, could become a lack of respect for their gender ascriptions.

Reviewer response: From viewing the submitted manuscript, I see that this section has now been deleted. Regardless, it is important to clarify this point. Not assuming gender wouldn’t ever result in the automatic use of he/him – if pronouns were ever essential to be used, the pronouns used generally are they/them when not assuming gender. In addition, you cannot be sure of a person’s gender from their first name – this bears no reflection of gender. I’m unsure what the authors mean when they say ‘their productions’. Generally, ‘the authors found’ or ‘the study found’ is used instead of gendered terms – this is the practice you have used elsewhere in this article.

7. PLOS authors have the option to publish the peer review history of their article (what does this mean?). If published, this will include your full peer review and any attached files.

Reviewer #2: **Yes: **Dr Rebecca A Jones

---

## [Author Response · Author response to Decision Letter 1]

13 Jun 2023

Dear Editor Professor Dr. Emily Chenette:

We are pleased to re-submit the revised version of the manuscript “Weight stigma after bariatric surgery: A qualitative study with Brazilian women”, as suggested. We have extensively revised the manuscript according to the suggestions made by the referee and the editor. An item-by-item response is presented below. All changes in the manuscript are highlighted with tracked changes. We hope that these changes will meet with your approval. 

- Editor:

Introduction

Editor comment: Some references need to be added/edited in the introduction section.

Response: We have corrected the references accordingly. 

Methods

Editor comment: In the research setting and study population section, the eligibility criteria includes gendered language (i.e., her). I assume men and women would be eligible for bariatric surgery, therefore referring to ‘her’ only is not appropriate.

Response: We have changed that as requested (page 7, lines 181, 182).

General comment

The full paper requires English language editing, engaging an editing service may support this. There are many long and complex sentences that need refining also.

Response: Three native English speakers have revised independently the text to address these questions. The changes regarding English language editing were not highlighted throughout the manuscript.

- Reviewer 1

Abstract

Original reviewer comment (R1): Can you please reframe the following into a research aim/objective (rather than a question): “However, how do bariatric patients perceive and internalize this stigma in different life stages? Specifically, is older age a risk factor?”. The current framing is a little too colloquial. Can you explicitly state the method of data analysis used? E.g., was it thematic analysis or something different? The abstract is missing a conclusion. Can you please add a sentence concluding your findings, and perhaps suggesting how the findings may be used in the future? You could make your results more concise if you need more word count to allow this.

Author response (R1): The changes in the Abstract were made accordingly.

Reviewer response (R2): You have not added the method of data analysis used to the abstract – please address this. The conclusion added goes beyond what your findings can support, and should be softened in tone to be more appropriate to the levels of findings. Specifically, the findings do not inform a targeted approach, however they might suggest future research should explore whether a targeted approach might be more effective.

Response (R2): The method of data analysis was informed as the following (page 2 lines 42-43): “The resulting text was then analyzed using thematic analysis”. To clarify, thematic analysis was the method used for data analysis. Regarding the conclusions, we have changed as suggested. In the previous version, it was as the following: “Our study can thus inform a more targeted approach to care for different cohorts who undergo bariatric surgery, an approach that would emphasize the importance of developing coping strategies with respect to experiences of stigma and discrimination after surgery”. We have changed in the revised version of the manuscript to (page 2, lines 53-56): “Our study suggest future research should explore whether a targeted approach might be more effective, for example, an approach that would emphasize the importance of developing coping strategies with respect to experiences of stigma and discrimination after surgery.” We have also added this information in other sections of the manuscript (i.e., at the end of Introduction and at the end of Conclusions).

Introduction

Original reviewer comment (R1): In the introduction, you use the stigmatising language of “fat”. Can you please address this throughout the manuscript to use person-first language?

Author response (R1): The change was made accordingly. We just kept the word “fat throughout the manuscript when it was an emic word or when referring to authors that employed this word.

Reviewer response (R2): Can you please edit the text to have quotation marks (or something similar) to make it clear to the reader than the remaining use of the word ‘fat’ is not your words, but quoting participants etc. 

Response (R2): We have added quotation marks to make it clear to the reader that the remaining use of the word ‘fat’ is not our words, but quoting participants or authors that employed this word.

Methods

Original reviewer comment (R1): Can you explain if and how you included patient and public involvement in this work? For example, what stages were they involved and what activities/input did they have.

Author response (R1): Participants were patients who underwent bariatric surgery at Hospital das Clínicas of the School of Medicine, University of São Paulo. Their involvement with the research included the individual semi-structured interviews. They were contacted by Author 1 and, if they agreed to participate, the interview was scheduled. These procedures are extensively described in the Methods section.

Reviewer response (R2): Patient and public involvement (PPI) is an active partnership between members of the public and researchers, where the public are involved in aspects of the study/research (such as informing the study design, supporting with analysis). The participants of the study do not count as PPI. You can read more about this on: https://www.rds-sc.nihr.ac.uk/ppi-information-resources/ Can you please provide a response given this explanation of PPI.

Response (R2): Thank you for your comment. We have not understood that this was the aim of your question, we apologize for that. We have clarified the Patient and Public involvement as the following (novel information is in bold), in the Methods section (page 8, lines 208-212): “In the initial contact, Author 1 introduced herself, explained why she was contacting them, the purpose of the research, and clarified that if they agreed to participate, the participation would be voluntary, and the findings would be kept confidential. This study does not qualify as Patient and Public Involvement (PPI) research, because PPI refers to an active partnership between members of the public and researchers, in which members of the public work alongside the research team and are actively involved in contributing to the research process as advisers and possibly as co-researchers. This did not happen in our study (NIHR, 2023). 

General comment

Original reviewer comment (R1): Line 631 – we do not tend to gender the authors of literature we are citing as we cannot assume one’s gender. Can you please review this throughout the manuscript.

Authors response (R1): We believe it is essential to point out that all the authors who sign this article stand out for investigating the different social markers of difference, including gender and sexuality, from an intersectional perspective. In this sense, the assumption of a particular gender is far from our perspective. If we de-gender an author, we would use the pronouns he and him, which are already gendered. Therefore, we decided to continue to refer to the authors as they refer to themselves as a gender through their productions and first names. We believe that de-gendering authors, without their consent, could become a lack of respect for their gender ascriptions.

Reviewer response (R2): From viewing the submitted manuscript, I see that this section has now been deleted. Regardless, it is important to clarify this point. Not assuming gender wouldn’t ever result in the automatic use of he/him – if pronouns were ever essential to be used, the pronouns used within English written language are they/them when not assuming gender. In addition, you cannot be sure of a person’s gender from their first name – this bears no reflection of gender. I’m unsure what the authors mean when they say ‘their productions’. Generally, ‘the authors found’ or ‘the study found’ is used instead of gendered terms – this is the practice you have used elsewhere in this article.

Response (R2): We thank you for your clarification. If the section was in the manuscript, we would certainly change it according to your recommendations. 

We thank you again for the valuable contributions of the reviewer and yourself and for the opportunity to submit our manuscript for your consideration.

Kind regards,

The authors.

---

## [Editor Report · Decision Letter 2]

15 Jun 2023

Weight stigma after bariatric surgery: A qualitative study with Brazilian women.

PONE-D-22-24534R2

Dear Dr. Dimitrov Ulian,

We’re pleased to inform you that your manuscript has been judged scientifically suitable for publication and will be formally accepted for publication once it meets all outstanding technical requirements.

Kind regards,

Vidanka Vasilevski

Academic Editor

PLOS ONE

Additional Editor Comments:

Rather than returning the paper for minor revision, I decided to accept the paper and ask you to make one minor change once you get to the copy edit stage. The first sentence in the introduction does not make sense: 

"One of the most effective means currently available to individuals seeking to lose weight deemed medically excessive is bariatric surgery." 

I believe the term excessive in the above sentence may need to be replaced with necessary?

Can you please rectify this when you review the copy edit of your paper prior to submission.

---

## [Editor Report · Acceptance letter]

19 Jul 2023

PONE-D-22-24534R2 

Weight stigma after bariatric surgery: A qualitative study with Brazilian women. 

Dear Dr. Dimitrov Ulian:

I'm pleased to inform you that your manuscript has been deemed suitable for publication in PLOS ONE. Congratulations! Your manuscript is now with our production department. 

Kind regards, 

on behalf of

Dr. Vidanka Vasilevski 

Academic Editor

PLOS ONE